# Cerebellar nuclei cells produce distinct pathogenic spike signatures in mouse models of ataxia, dystonia, and tremor

**Meike E van der Heijden**[1,2]*[†‡], **Amanda M Brown**[1,2][†], **Dominic J Kizek**[2], **Roy V Sillitoe**[1,2,3,4,5]*

[1]Department of Pathology & Immunology, Baylor College of Medicine, Houston, United States; [2]Jan and Dan Duncan Neurological Research Institute at Texas Children's Hospital, Houston, United States; [3]Department of Pediatrics, Baylor College of Medicine, Houston, United States; [4]Development, Disease Models & Therapeutics Graduate Program, Baylor College of Medicine, Houston, United States; [5]Department of Neuroscience, Baylor College of Medicine, Houston, United States

**\*For correspondence:**
mheijden@vtc.vt.edu (MEvdH);
sillitoe@bcm.edu (RVS)

[†]These authors contributed equally to this work

**Present address:** [‡]Fralin Biomedical Research Institute, Virginia Tech Carilion, Roanoke, United States

## Abstract

The cerebellum contributes to a diverse array of motor conditions, including ataxia, dystonia, and tremor. The neural substrates that encode this diversity are unclear. Here, we tested whether the neural spike activity of cerebellar output neurons is distinct between movement disorders with different impairments, generalizable across movement disorders with similar impairments, and capable of causing distinct movement impairments. Using in vivo awake recordings as input data, we trained a supervised classifier model to differentiate the spike parameters between mouse models for ataxia, dystonia, and tremor. The classifier model correctly assigned mouse phenotypes based on single-neuron signatures. Spike signatures were shared across etiologically distinct but phenotypically similar disease models. Mimicking these pathophysiological spike signatures with optogenetics induced the predicted motor impairments in otherwise healthy mice. These data show that distinct spike signatures promote the behavioral presentation of cerebellar diseases.

## eLife assessment

The authors' dataset and analysis provide a **fundamental** new understanding of how cerebellar output contributes to various cerebellar-dependent diseases. The observation that different firing statistics at the level of the cerebellar nuclei directly impart disease-specific phenotypes is quite **convincing**. The classifier used in the article remains a potential weak point, showing limited efficacy, particularly for identifying mice with tremor. The concern about classifier accuracy is ameliorated by the fact that the classifier parameters are easily interpretable, and allowed the authors to use these parameters to design stimulation experiments.

## Introduction

There exists a historical and rich curiosity in understanding the role of the cerebellum in movement, dating back to the pioneering work of *Flourens, 1841*, with an equally long interest in investigating how it alters movement (*Holmes, 1917*; *Turner, 1892*). From these earlier studies, it was clear that a defective cerebellum causes a range of devastating problems in the ability to control voluntary, intentional actions, including coordination, posture, and balance. Accordingly, in humans, the equivalent defects had clear pathophysiological consequences. Patients with various cerebellar lesions showed

**eLife digest** Intentional movement is fundamental to achieving many goals, whether they are as complicated as driving a car or as routine as feeding ourselves with a spoon. The cerebellum is a key brain area for coordinating such movement. Damage to this region can cause various movement disorders: ataxia (uncoordinated movement); dystonia (uncontrolled muscle contractions); and tremor (involuntary and rhythmic shaking).

While abnormal electrical activity in the brain associated with movement disorders has been recorded for decades, previous studies often explored one movement disorder at a time. Therefore, it remained unclear whether the underlying brain activity is similar across movement disorders.

Van der Heijden and Brown et al. analyzed recordings of neuron activity in the cerebellum of mice with movement disorders to create an activity profile for each disorder. The researchers then used machine learning to generate a classifier that could separate profiles associated with manifestations of ataxia, dystonia, and tremor based on unique features of their neural activity. The ability of the model to separate the three types of movement disorders indicates that abnormal movements can be distinguished based on neural activity patterns.

When additional manifestations of these abnormal movements were considered, multiple mouse models of dystonia and tremor tended to show similar profiles. Ataxia models had several different types of neural activity that were all distinct from the dystonia and tremor profiles. After identifying the activity associated with each movement disorder, Van der Heijden and Brown et al. induced the same activity in the cerebella of healthy mice, which then caused the corresponding abnormal movements.

These findings lay an important groundwork for the development of treatments for neurological disorders involving ataxia, dystonia, and tremor. They identify the cerebellum, and specific patterns of activity within it, as potential therapeutic targets. While the different activity profiles of ataxia may require more consideration, the neural activity associated with dystonia and tremor appears to be generalizable across multiple manifestations, suggesting potential treatments could be broadly applicable for these disorders.

loss of precise motor coordination, which, as the consequence of disease, is referred to as ataxia, whereas some subjects displayed overt and sometimes exaggerated oscillatory movements, consistent with tremor (*Holmes, 1917*; *Turner, 1892*; *Holmes, 1939*). In other cases, the examiner would report uncontrolled muscle contractions, which is now a contributing feature of dystonia (*Balint et al., 2018*). These classic observations underscored the heterogeneity of motor disturbances caused by cerebellar dysfunction. But, even then, the question of how the cerebellum creates such behavioral diversity was already imminent. Despite major advances in understanding the basic anatomy and circuitry of the cerebellum, it is still unclear how disease behaviors emerge from these circuits. In this regard, the specific substrates that underlie each disorder could hold the key for improving the design of therapies and treatments.

Consistent with the outcomes of removing or lesioning the cerebellum in pigeons, dogs, monkeys, and humans (*Holmes, 1917*; *Turner, 1892*; *Holmes, 1939*), the heterogeneity of cerebellar movement disturbances was later confirmed in genetic mouse models. Much of the current excitement about the cerebellum spawned from these initial genetic models because of the overt motor disturbances that were caused by spontaneous mutations in genes that are now known to be associated with cerebellar development or cerebellar degeneration (*Falconer, 1951*; *Sidman et al., 1965*). Prior to genetic sequencing though, mutant mice such as *hot-foot* (*Guastavino et al., 1990*), *weaver* (*Rakic and Sidman, 1973*), *trembler* (*Falconer, 1951*), *waddles* (*Jiao et al., 2005*), *staggerer* (*Sidman et al., 1962*), *stumbler* (*Caddy et al., 1981*), *tottering* (*Green and Sidman, 1962*), and *lurcher* (*Phillips, 1960*) were named according to phenotype of their abnormal movements, which are as diverse as their names imply. Modern techniques and approaches now aim at determining the mechanisms and roles of the cerebellum as they relate to driving different behaviors in these classic models (*Snell et al., 2022*; *Pan et al., 2020*). Together, data generated from these different mouse models have cultivated an interest in identifying whether discrete functional features in the cerebellar circuit are the root cause of disease-related behaviors. In this regard, more than half a century since the first descriptions of cerebellar mutants, a core question

remains unanswered but hotly debated: how does cerebellar circuit dysfunction lead to unique motor disturbances?

To begin addressing how cerebellar circuits generate behavioral diversity in disease, we used an intersectional genetics approach to mark, map, and manipulate specific types of synapses in the cerebellum. Our approach silences genetically defined populations of synapses by selectively deleting the genes that encode the vesicular GABA transporter, SLC32A1 (also known as VGAT), or the type 2 vesicular glutamate transporter, SLC17A6 (also known as VGLUT2), from genetically targeted populations of cells. Loss of GABAergic neurotransmission from Purkinje cells, which provide the sole output of the cerebellar cortex, caused uncoordinated movements and disequilibrium that were indicative of ataxia (*Figure 1A*; *White et al., 2014*). In contrast, eliminating glutamatergic neurotransmission from climbing fibers, which synapse onto Purkinje cells, caused twisting postures and hyperextended limbs that are consistent with dystonia (*Figure 1A*; *White and Sillitoe, 2017*). In addition, systemic injection of the β-carboline alkaloid drug harmaline caused hyperactivation of climbing fibers and rhythmic bursting spike activity in Purkinje cells as well as a severe tremor (*Figure 1A*). Optogenetically modulating Purkinje cell projections to cerebellar nuclei cells in a bursting pattern induced oscillatory tremor movements (*Stratton and Lorden, 1991*; *Brown et al., 2020b*). Together, these studies established that different manipulations of Purkinje cell inputs or outputs, and consequently Purkinje cell and nuclei neuron spike signals, can cause diverse behavioral deficits as they relate to disease (*White and Sillitoe, 2017*; *Brown et al., 2020b*; *van der Heijden and Sillitoe, 2023*). Given the heterogeneity and even comorbidity of these behaviors in a single disease model (*White et al., 2016a*), the main question that arises is, do these cerebellar neural signals represent unique pathophysiological signatures that can drive the predominant behavioral defects used to characterize different motor diseases? Here, we aim to resolve this question to provide insight into the origin of cerebellar movement disorder presentation.

## Results

### Spike signatures are different between archetypal ataxia, dystonia, and tremor models

We first compared the spike train activity between mouse models for ataxia (*Pcp2^Cre^;Slc32a1^fl/fl^*, *White et al., 2014*; *Brown et al., 2020b*), dystonia (*Ptf1a^Cre^;Slc17a6^fl/fl^*, *White and Sillitoe, 2017*; *Brown et al., 2023*), and tremor (harmaline injection, *Brown et al., 2020b*; *Handforth, 2012*; *Figure 1A*). We identified these three mouse models as archetypal representations of their respective cerebellar motor disease phenotypes because their overt behavioral motor abnormalities are caused by cerebellum-specific manipulations that do not cause changes in the gross anatomy, cell morphology, or cell survival of the adult cerebellum. These models also exhibit remarkably reliable, severe, and penetrant ataxic, dystonic, and tremor symptoms within each respective group. *Video 1* provides examples of the varied and overt motor abnormalities of the mouse models used. *Supplementary file 1* summarizes the reported behavioral impairments from prior characterization of these archetypal mouse models.

We hypothesized that these differences in behavioral abnormalities were accompanied by specific changes in the spike train patterns in the cerebellum. We analyzed in vivo electrophysiology recordings of the spike activity in cerebellar nuclei neurons of the interposed nucleus (*Figure 1B and C*), in awake, head-fixed mice with overt motor phenotypes and control animals. We focused our recordings on the interposed nucleus based on the hypothesis that the cerebellum communicates ongoing motor signals to other brain regions through this nucleus (*Low et al., 2018*) and deep brain stimulation (DBS) of this region successfully reduces motor impairments in mouse models of ataxia (*Miterko et al., 2021*), dystonia (*White and Sillitoe, 2017*), and tremor (*Brown et al., 2020b*). We described the spike train firing features using 12 parameters that summarized the spike train rate (*Figure 1— figure supplement 1A–C*), irregularity (*Figure 1—figure supplement 1D–F*), pauses (*Figure 1— figure supplement 1G–I*), and rhythmicity (*Figure 1—figure supplement 1J–L*). In agreement with our hypothesis, we found a significant difference between at least two archetypal groups for each of the 12 spike train parameters. However, none of the 12 parameters showed a statistical difference between all four groups (control, ataxia, dystonia, and tremor), suggesting that the difference between spike train signatures relied on a combination of multiple spike train parameters.

**Figure 1.** Supervised classifier model predicts mouse phenotype based on spike signatures. (**A**) Schematic of network changes causing motor impairments in mouse models for ataxia, dystonia, and tremor. Dotted lines indicate lack of neurotransmission. Color-coded lines indicate primary affected cell type. (**B**) Example of spike firing rate averaged over previous 50 ms at each occurring spike for the 5 s spike train in (**B′**). (**B″**) 1 s spike train for the duration indicated in the square box in (**B′**). (**C**) Histograms of instantaneous firing rate (ISI⁻¹) of the full 30 s spike train used in the classification

*Figure 1 continued on next page*

*Figure 1 continued*

model. We indicate the firing rate (spikes/s), median ISI$^{-1}$, skewness, and ISI$_{>25}$ for all for example cells. (**D**) Classifier model based on training data set (control: n = 25 cells; ataxia: n = 20; dystonia: n = 20; tremor: n = 20). (**E**) Assigned spike signature based on spike properties in complete data set (control: n = 33 cells, N = 9 mice; ataxia: n = 24, N = 5; dystonia: n = 24, N = 9; tremor: n = 24, N = 6). Categories on x-axis indicate the origin of the recorded neurons. (**F, G**) Scatterplot of spike train parameters used to classify neural signatures. Colored boxes indicate the predicted phenotype, colors of circles indicate the origin of the recorded neurons. PC = Purkinje cell; CN = cerebellar nuclei; IO/CF = inferior olive/climbing fiber.

The online version of this article includes the following source data and figure supplement(s) for figure 1:

**Source data 1.** Source data for the graphs in *Figure 1* and its figure supplements.

**Figure supplement 1.** Distribution of spike train parameters in control, ataxic, dystonic, and tremor mice.

**Figure supplement 2.** Correlation coefficients for spike parameters in control, ataxic, dystonic, and tremor mice.

**Figure supplement 3.** Classifier models based on 12 different sets of training data.

Even though each parameter describes a different feature of the spike train activity, there is a strong correlation between many parameter values in each group of recordings (*Figure 1—figure supplement 2*). We therefore set out to find the minimal set of parameters and specific threshold values that could differentiate the spike train parameters between the archetypal groups. We trained 12 supervised classification learning models based on the spike train parameters of each mouse model (*Figure 1—figure supplement 1A–L*) with different sets of cells used for training and validation (*Figure 1—figure supplement 3*). We sought to determine which classifier model best assigned the validation neurons' spike signatures to the correct origin mouse model. We found that the two classifier models (models 10 and 9) with the best performance in the validation set used the same parameters and parameter cutoffs to classify the spike signatures between control, ataxia, dystonia, and tremor mice (*Figure 1—figure supplement 3*). We therefore used this classifier throughout the remainder of the article to assign a phenotypic signature to the spike train properties of cerebellar nuclei neurons.

The classifier model identified 'skewness' (*Figure 1D*) as the best differentiator between neurons recorded in ataxic and control mice (lower skewness) and neurons recorded in dystonic and tremoring mice (higher skewness). Skewness is a measure of the relative skew in the interspike interval (ISI) distribution of single neurons from the median to the mean (*Figure 1—figure supplement 1F*) and is unusually elevated in our dystonic and tremor mouse models (*Figure 1C*). Next, the classifier model identified CV2 (*Holt et al., 1996*; *Figure 1—figure supplement 1E*) as the best differentiator of neurons recorded in ataxic mice (lower CV2) from control mice (higher CV2). CV2 measures the irregularity of ISIs with less influence from the overall firing rate. It is extraordinarily low in the highly regular spike activity observed in our ataxia model (*Figure 1B″*). Last, the classifier model identified that the ISI$_{>25}$ (*Figure 1—figure supplement 1C*), which measures the percentage of ISI with a duration over 25 ms (or instantaneous firing rate under 40 Hz) (*Figure 1—figure supplement 1G*), as the best differentiator of neurons from the mice with tremor (lower ISI$_{>25}$) compared to those recorded in dystonic mice (higher ISI$_{>25}$) (*Figure 1C*). These findings suggest that changes in a combination of three parameters (skewness, CV2, and ISI$_{>25}$) can differentiate neural spike signatures in mice with distinct behavioral presentation of cerebellar disease.

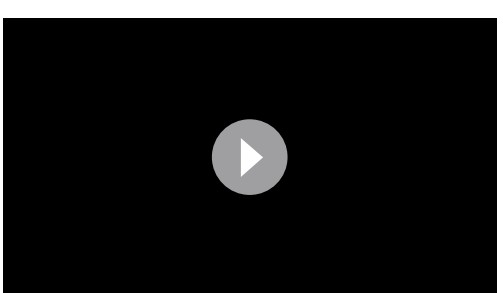

**Video 1.** Examples of freely moving mouse behavior in a control mouse, an ataxic mouse, a dystonic mouse, and a tremoring mouse.
https://elifesciences.org/articles/91483/figures#video1

We assessed whether this classifier model correctly assigns the spike signature corresponding to the behavioral phenotypes of the archetypal mouse models based solely on the spiking activity of single cerebellar nuclei neurons. We tested this by assigning a signature based on skewness, CV2, and ISI$_{>25}$ values. We found that the classifier model indeed assigned the signature that aligned with different predominant phenotypes of each archetypal mouse model correctly for the majority of neurons (control: 85% of neurons recorded in a control mouse had a control signature; ataxia: 79%; dystonia: 88%; tremor: 54%) (*Figure 1E*). Our analyses also

van der Heijden, Brown *et al*. eLife 2023;12:RP91483. DOI: https://doi.org/10.7554/eLife.91483

showed that most neurons of a specific signature were recorded from the mouse model with that phenotype (control: 63% of neurons with a control signature were recorded in a control mouse; ataxia: 93%; dystonia: 74%; tremor: 88%) (*Figure 1E–G*). Furthermore, the relative proportion of neurons with a control signature was significantly enriched in the control population relative to the entire population (chi-square: p<0.0001) (*Figure 1E*), and we also found this enrichment for the ataxia signature in the ataxia neurons (p<0.0001), dystonia signature in the dystonia neurons (p<0.0001), and tremor signature in the tremor neurons (p<0.0001). These data support the hypothesis that spike signatures are reliably different between archetypal mouse models of distinct motor behaviors that mimic human disease symptoms.

## Multiple spike train signatures can lead to ataxic motor impairments

Next, we set out to identify whether the spike train signatures for the archetypal mouse models are shared across mouse models with different manipulations but similar phenotypes. We started by classifying the spike train parameters in a mouse model of ataxia caused by a poly-glutamate expansion in the Atxn1 gene, $Atxn1^{154Q/+}$ mice, that causes spinocerebellar ataxia type 1 (SCA1). We measured the spike train patterns early in the disease progression (*Coffin et al., 2023*; *Figure 2A and B*). Despite the presence of an ataxic phenotype at this age (*Supplementary file 1*), we observed that most $Atxn1^{154Q/+}$ neurons exhibited a control signature (22/34) and that the second most numerous spike train signature was a dystonia signature (10/24).

We reasoned that the high proportion of cells exhibiting a control signature in $Atxn1^{154Q/+}$ mice might be because the disease-causing genetic abnormality affects the whole body and, consequently, the ataxic symptoms may result from dysfunction in multiple nodes in the motor circuit. We therefore also investigated a mouse model with a cerebellum-specific loss of the gene *Ank1* (10 to 12-month-old $Pcp2^{Cre};Ank1^{fl/fl}$ mice, *Figure 2A*), which causes an adult-onset degenerative ataxia (*Stevens et al., 2022*). In this mouse model, we found that half of the neurons exhibited an abnormal spike train signature, with the majority classified with the dystonia spike train signature (6/14).

Our classifier model differentiates the control and dystonia signatures from each other based on the skewness parameter. Previous studies have suggested that the difference between ataxia and dystonia symptoms may be caused by the relative difference in ISI regularity (*Snell et al., 2022*; *Shakkottai et al., 2017*; *Shakkottai, 2014*), which can be measured by our skewness parameter. Indeed, the population distribution of skewness values was lower in the ataxic $Atxn1^{154Q/+}$ (0.14 ± 0.02 [mean ± SEM]) and $Pcp2^{Cre};Ank1^{fl/fl}$ (0.17 ± 0.03) mouse models (*Figure 2D and E*) than in our archetypal dystonia model (0.42 ± 0.05) ($Ptf1a^{Cre};Slc17a6^{fl/fl}$, *Figure 1F and G*). We used a one-way ANOVA followed by Tukey post hoc analyses and found that the difference in skewness between the ataxia models and the dystonia models was statistically significant ($Atxn1^{154Q/+}$ vs. $Ptf1a^{Cre};Slc17a6^{fl/fl}$: p<0.0001; $Pcp2^{Cre};Ank1^{fl/fl}$ vs. $Ptf1a^{Cre};Slc17a6^{fl/fl}$: p = 0.0025), but we did not observe a difference in skewness between the two ataxia models ($Atxn1^{154Q/+}$ vs. $Pcp2^{Cre};Ank1^{fl/fl}$: p = 0.9961). Together, these findings suggest that there are multiple cerebellar spike train signatures that can lead to ataxic symptoms that both differ from the dystonia signature in their relative level of regularity; unusually regular spike train patterns are seen in $Pcp2^{Cre};Slc32a1^{fl/fl}$ mice (*Figure 1*) and intermediate irregular spike train patterns are observed in $Atxn1^{154Q/+}$ and $Pcp2^{Cre};Ank1^{fl/fl}$ mouse models (*Figure 2*).

## Dystonic spike train signatures are shared across etiologically distinct dystonia models

Next, we set out to identify whether the dystonia signature was shared across mice with overt dystonic phenotypes. We included two dystonia models in our analyses; $Pdx1^{Cre};Slc17a6^{fl/fl}$ mice exhibit mild dystonic symptoms only in response to a stressful event (*Lackey, 2021*; *Leon and Sillitoe, 2023*; *Figure 3A and B*), whereas mice that receive ouabain applied to the surface of the cerebellum exhibit continuous severe dystonic features (*Fremont et al., 2014*; *Calderon et al., 2011*; *Figure 3A and B*). We found that the proportion of neurons that exhibited the control signature was greater in $Pdx1^{Cre};Slc17a6^{fl/fl}$ mice than in ouabain infusion mice (7/23 vs. 13/21, z = −2.094, p = 0.0366; *Figure 3C*). Moreover, the proportion of neurons with a dystonia signature was lower in $Pdx1^{Cre};Slc17a6^{fl/fl}$ mice than in ouabain infusion mice (15/23 vs. 7/21, z = 2.1128, p = 0.0349; *Figure 3C*). Therefore, the mice with a more severe dystonia phenotype had a smaller proportion of cells matching the control signature and a greater proportion of cells matching the dystonia signature. In addition to the proportional

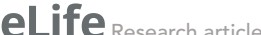

**Figure 2.** Spike signatures in ataxia models with different etiologies. (**A**) 5 s example spike trains (and 1 s inset) of representative cerebellar nuclei neurons recorded in each mutant mouse model. Left: example cell from early disease progression (4-month-old) ataxic *Atxn1*$^{154Q/+}$ mouse. Right: example cell from late disease progression (11-month-old) ataxic *Pcp2*$^{Cre}$;*Ank1*$^{fl/fl}$ mouse. (**B**) Histograms of instantaneous firing rate (ISI$^{-1}$) of the full 30 s spike train of the example cells in (**A**). We indicate the firing rate (spikes/s), median ISI$^{-1}$, skewness, and ISI$_{>25}$. (**C**) Proportion of predicted spike signatures in each of the mouse models. (**D, E**) Scatterplot of spike

*Figure 2 continued on next page*

*Figure 2 continued*

train parameters used to classify neural signatures. Colored boxes indicate the predicted phenotype, colors of circles indicate the origin of the recorded neurons. Control: n = 27 cells, N = 5 mice; *Atxn1*$^{154Q/+}$: n = 34, N = 4; *Pcp2*$^{Cre}$;*Ank1*$^{fl/fl}$: n = 14, N = 3. (**C–E**) are based on the classifier model in *Figure 1D*.

The online version of this article includes the following source data for figure 2:

**Source data 1.** Source data for the graphs in *Figure 2*.

differences in cells with a control and dystonia signature, we found that the distribution of skewness values differed between the *Pdx1*$^{Cre}$;*Slc17a6*$^{fl/fl}$ (0.12 ± 0.05) and ouabain infusion mice (0.36 ± 0.8) (*Figure 3D and E*). We quantified this difference by comparing these two mouse models to each other and the archetypal dystonic model (*Ptf1a*$^{Cre}$;*Slc17a6*$^{fl/fl}$) using a one-way ANOVA followed by Tukey post hoc analysis. We found that the skewness was lower in *Pdx1*$^{Cre}$;*Slc17a6*$^{fl/fl}$ mice with mild dystonic features compared to the ouabain infusion mice (p = 0.0012) and *Ptf1a*$^{Cre}$;*Slc17a6*$^{fl/fl}$ mice (p<0.0001) with severe dystonic features. There was no significant difference between the two models with severe dystonic features (p = 0.9274). These findings show that the dystonia signature is shared across etiologically distinct dystonia models and that the relative severity of symptoms may be explained by the proportion of neurons with a dystonia and control signature (*Figure 3*).

## Tremor spike train signatures are shared across etiologically distinct tremor models

Next, we set out to identify whether the tremor signature is shared across mice that exhibit severe oscillatory tremors. To test this, we used *Car8*$^{wdl/wdl}$ mice. We previously showed that *Car8*$^{wdl/wdl}$ mice exhibit a severe oscillatory tremor that can be reduced to control levels by treatment with the β-blocker drug propranolol (*Zhou et al., 2022*), which is used to treat tremor in human patients. *Car8*$^{wdl/wdl}$ mice also exhibit ataxia and dystonia, which are not normalized by propranolol (*Jiao et al., 2005*; *White et al., 2016a*; *Miterko et al., 2021*; *Zhou et al., 2022*). We found that, before treatment, *Car8*$^{wdl/wdl}$ mice exhibited a large proportion of neurons with the tremor signature (8/13). However, in propranolol-treated *Car8*$^{wdl/wdl}$ mice the proportion of neurons with the tremor signature was diminished while control (6/15) and dystonia (6/15) signatures were the most common signatures (*Figure 4C*). The classifier model differentiates between the dystonia and tremor signature based on the ISI$_{>25}$ value, and, indeed, we find a shift toward higher ISI$_{>25}$ values in propranolol-treated *Car8*$^{wdl/wdl}$ mice compared to untreated *Car8*$^{wdl/wdl}$ mice (*Figure 4D and E*). Together, these findings suggest that the tremor signature is shared across etiologically distinct mouse models of tremor and that the tremor-reducing drug propranolol also reduces specific spike train features contributing to the tremor signature (*Figure 4*).

## Different spike signatures can be generated by the same canonical cerebellar circuit

Thus far, the observed spike train signatures have been dependent on developmental changes in circuit connectivity (*Pcp2*$^{Cre}$;*Slc32a1*$^{fl/fl}$; *Ptf1a*$^{Cre}$;*Slc17a6*$^{fl/fl}$; *Pdx1*$^{Cre}$;*Slc17a6*$^{fl/fl}$), neurodegeneration (*Atxn1*$^{154Q/+}$; *Pcp2*$^{Cre}$;*Ank1*$^{fl/fl}$), the effects of transgene expression (*Atxn1*$^{154Q/+}$) or frameshifting (*Car8*$^{wdl/wdl}$), or drug effects on the nervous system outside of the cerebellum (harmaline, ouabain, propranolol). Therefore, to determine whether the distinctive features of spike activity could be generated by the same, otherwise healthy circuit, we expressed a light-sensitive cation channel opsin, channelrhodopsin (ChR2), in Purkinje cells (*Pcp2*$^{Cre}$;*ROSA26*$^{loxP-STOP-loxP-EYFP-ChR2}$ mice, hereon referred to as *Pcp2*$^{Cre}$;*ROSA26*$^{ChR2}$ mice). Upon light activation, Purkinje cell firing rate increases (*Figure 5—figure supplement 1*) and nuclei neurons are inhibited via the GABAergic neurotransmission from Purkinje cells (*Figure 5A–C*). We leveraged this circuit connectivity and optogenetic strategy to stimulate Purkinje cell terminals in the interposed nucleus to induce distinct spike signatures in those downstream interposed nucleus neurons (*Figure 5*). We chose three stimulation paradigms. The ataxia stimulation was a continuous 50 Hz square pulse stimulation (mimicking the lack of modulation by Purkinje cells in our degenerative and genetically silenced ataxia models); the dystonia stimulation was a 50 Hz square pulse stimulation that was on for at least 1 s with a 75% chance of a 250 ms pause between periods of stimulation (to induce the slow and irregular spike train features of our defined dystonia signature); and the tremor stimulation was a 10 Hz sinusoidal stimulation (to induce the fast, irregular,



**Figure 3.** Spike signatures in dystonia models with different etiologies. (**A**) 5 s example spike trains (and 1 s inset) of representative cerebellar nuclei neurons recorded in each mutant mouse model. Left: example cell from mildly dystonic *Pdx1^{Cre};Slc17a6^{fl/fl}* mouse. Right: example cell from severely dystonic cerebellum ouabain infusion mouse. (**B**) Histograms of instantaneous firing rate (ISI$^{-1}$) of the full 30 s spike train of the example cells in (**A**). We indicate the firing rate (spikes/s), median ISI$^{-1}$, skewness, and ISI$_{>25}$. (**C**) Proportion of predicted spike signatures in each of the mouse models. (**D, E**) Scatterplot of spike train parameters used to classify neural signatures. Colored boxes

*Figure 3 continued on next page*

*Figure 3 continued*

indicate the predicted phenotype, colors of circles indicate the origin of the recorded neurons. Control: n = 28 cells, N = 6 mice; *Pdx1*$^{Cre}$;*Slc17a6*$^{fl/fl}$: n = 23, N = 5; cerebellum ouabain infusion: n = 21, N = 4. (**C–E**) are based on the classifier model in *Figure 1D*.

The online version of this article includes the following source data for figure 3:

**Source data 1.** Source data for the graphs in *Figure 3*.

and rhythmic spike train features of our defined tremor signature). We performed in vivo recordings in awake, head-fixed mice, and confirmed that each of the optogenetic stimulation paradigms caused distinct changes in Purkinje cell spiking activity (*Figure 5—figure supplement 1*). We then directed stimulation toward Purkinje cell terminals surrounding interposed nucleus neurons. The stimulation paradigms induced distinct changes in the spike train features of interposed nuclei neurons using this method (*Figure 5D*). We then compared the resultant spike train features to the defined thresholds of our classifier model. We found that the ataxia stimulation paradigm induced cells to develop a response similar to the *Atxn1*$^{154Q/+}$ and *Pcp2*$^{Cre}$;*Ank1*$^{fl/fl}$ ataxia model mice (*Figure 3*, *Figure 5—figure supplement 2*), with about half of the cells remaining within the control signature (7/16), and half shifting into the dystonia signature (8/16) (*Figure 5F*). Also akin to these models, when we used a one-way ANOVA followed by Tukey post hoc analysis to compare the ataxia and dystonia stimulation responses, we found the ataxia stimulation resulted in significantly lower skewness than the dystonia stimulation (ataxia stimulation versus dystonia stimulation: p = 0.0287). Additionally, we found that the dystonia and tremor stimulation induced the majority of cells to produce spike signatures that our classifier model defined as dystonic (9/16 cells) or as tremor signatures (11/16 cells), respectively (*Figure 5F*). These data indicate that spike signatures associated with ataxia, dystonia, and tremor can be generated by the same, otherwise healthy, cerebellar circuit.

## Spike signatures can induce distinct motor phenotypes that mimic disease-related behaviors

During the head-fixed stimulation experiments, we observed subtle behavioral responses during unilateral cell-targeted optogenetic stimulation that suggested the different spike signatures might drive unique motor behaviors (*Video 2*). To explore this further in freely moving mice, we investigated whether the different spike signatures directly drive unique motor disturbances when a population of cells are induced to produce our defined spike signatures. To this end, *Pcp2*$^{Cre}$;*ROSA26*$^{ChR2}$ mice were implanted with optical fibers that were bilaterally targeted to the interposed cerebellar nuclei (*Figure 6A–C*). This allowed for the targeting of Purkinje cell terminals in the cerebellar nuclei, as was done during our in vivo recordings in *Figure 5*, but with a larger population of cells affected by the stimulation. This also allowed the mice to move freely during stimulation, making a series of behavioral assays during stimulation possible (*Figure 6D*). Changes in behavior were immediate and severe as soon as the stimulation began. Each stimulation paradigm resulted in a different constellation of behaviors, with the ataxia stimulation paradigm resulting in imprecise movements, the dystonia stimulation paradigm resulting in prominent dystonic posturing, and the tremor stimulation paradigm resulting in severe tremor (*Video 3*). Therefore, we assessed the mice for changes in gait, presence of dystonic movements, and severity of tremor. No single measurement can perfectly encapsulate the complexity of ataxia, dystonia, and tremor – all of which can appear in combination with the others (*Pandey and Sarma, 2016*; *Nibbeling et al., 2017*; *Hagerman and Hagerman, 2015*). Additionally, each phenotype may affect the measurement of others. This was particularly apparent with measurements of gait. We found that all stimulation paradigms affected gait, resulting in visibly different foot placement (*Figure 6E-H*) and movement down the footprinting corridor (*Video 4*). Quantitatively, this was evident in shorter steps and less precision of fore and hindpaw placement (*Figure 6I–K*). We also considered the behavior of the mice in an open, flat, plexiglass arena for signs of dystonic movements (*Figure 6L*, *Video 3*). The dystonia stimulation paradigm resulted in the most frequent and strongest dystonic movements, while we also observed dystonic movements with the ataxia paradigm and abnormal movement – namely severe tremor – was noted with the tremor stimulation paradigm (*Figure 6M*). Mice were also placed in a custom-made tremor monitor (*Brown et al., 2020b*) where they could freely ambulate during stimulation while an accelerometer mounted under the arena

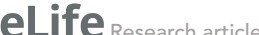

**Figure 4.** Spike signatures in tremor models with different etiologies. (**A**) 5 s example spike trains (and 1 s inset) of representative cerebellar nuclei neurons recorded in each mutant mouse model. Left: example cell from $Car8^{wdl/wdl}$ mouse with complex phenotype including severe tremor. Right: example cell from $Car8^{wdl/wdl}$ mouse with complex phenotype treated with propranolol to treat tremor. (**B**) Histograms of instantaneous firing rate ($ISI^{-1}$) of the full 30 s spike train of the example cells in (**A**). We indicate the firing rate (spikes/s), median $ISI^{-1}$, skewness, and $ISI_{>25}$. (**C**) Proportion of predicted spike signatures in each of the mouse models. (**D, E**) Scatterplot of spike train

*Figure 4 continued on next page*

*Figure 4 continued*

parameters used to classify neural signatures. Colored boxes indicate the predicted phenotype, colors of circles indicate the origin of the recorded neurons. Control: n = 14 cells, N = 4 mice; *Car8*$^{wdl/wdl}$: n = 13, N = 6; *Car8*$^{wdl/wdl}$ + propranolol: n = 15, N = 6. (**C–E**) are based on the classifier model in *Figure 1D*.

The online version of this article includes the following source data for figure 4:

**Source data 1.** Source data for the graphs in *Figure 4*.

detected changes in acceleration (*Figure 6D*). While the ataxia stimulation paradigm did not significantly increase tremor from baseline, both the dystonia and tremor stimulation paradigms resulted in a significantly increased tremor at 10 Hz (*Figure 6N–P*). Though dystonia is often observed with tremor, it is possible that the detection of tremor in these animals was due to the jerkiness of their dystonic movements rather than an increase in smooth, rhythmic tremor that is more often associated with tremor disorders. Indeed, the tremor paradigm resulted in the most severe tremor of all stimulation conditions, producing a tremor severity that was more than tenfold greater than that of the dystonia stimulation paradigm. Together, these measurements produced a complex representation of the underlying phenotypes. All three stimulation parameters resulted in a behavioral repertoire that was significantly different from baseline and, while there was some overlap of features, each stimulation paradigm produced the distinct and predicted respective motor phenotype (*Figure 6Q–T*). Together these data suggest that our classifier model's defined spike signatures of disease-associated cerebellar nuclei spike trains are sufficient to produce the predicted corresponding abnormal motor phenotypes in mice.

## Discussion

In this study, we tested whether distinct spike train signatures in the interposed cerebellar nuclei explain why cerebellar dysfunction can cause multiple distinct motor impairments associated with movement disorders. By comparing spike activity across multiple mouse models of cerebellar disease, we found that the cerebellum can generate a range of dysfunctional spiking patterns. We found disease-specific spike train signatures using a classifier learner model, which allowed us to discover specific spike train parameters and their corresponding cutoff values that could distinguish the activity associated with these different disease states. When investigating whether these spike signatures are shared across mouse models with similar phenotypes due to different etiologies, we found that two types of spike train activity can cause ataxia, whereas specific spike train signatures are strongly associated with dystonia and tremor. We then tested whether we could optogenetically induce these signatures in an otherwise normal circuit. We found that the same neurons can generate healthy spike activity as well as a spectrum of disease-like spike activities. We then tested whether these optogenetically induced disease-associated spike signatures could elicit similar behavioral abnormalities in the absence of any other primary genetic or circuit defects in awake and freely moving mice. The predominant behaviors that characterize each disease condition were successfully recapitulated. These data provide compelling evidence for the reliance of neurological phenotype presentation on the pattern of cerebellar circuit misfiring.

Several previous studies have proposed that distinct spike train patterns may correspond to distinct presentations of cerebellar disease (*Snell et al., 2022*; *Shakkottai et al., 2017*; *Shakkottai, 2014*; *Tara et al., 2018*). Our work builds on these studies by quantitatively comparing spike train properties across, rather than within, mouse models. We demonstrate which aspects of the spike train patterns are distinct and shared across mouse models with different and similar disease phenotypes, respectively. We confirm prior research indicating the importance of spike train irregularity in disease presentation (*Snell et al., 2022*; *Shakkottai et al., 2017*; *Tara et al., 2018*). We also show that spike train irregularity is insufficient to differentiate the spike train properties of dystonic and tremoring mice, which are differentiated from each other based on the instantaneous firing rate rather than pattern. Additionally, we confirm that the spike train signatures associated with different disorders can cause distinct disease phenotypes, thereby showing for the first time that distinct cerebellar spike train signatures are sufficient to drive a variety of motor impairments. Together, these data provide strong evidence that different spike train signatures do not only result from sensory feedback toward the



**Figure 5.** Spike signatures in cerebellar nuclei neurons can be induced by specific stimulation paradigms of Purkinje cells. (**A**) Schematic of experimental setup with recordings in awake, head-fixed mice. (**B**) Optopatcher recordings of cerebellar nuclei neurons. The opsin is expressed in Purkinje cells (pink) and recordings of nuclei neurons are performed. PC = Purkinje cell; CN = cerebellar nuclei; IO/CF = inferior olive/climbing fiber. (**C**) Example validation that light stimulation of inhibitory Purkinje cells (blue bars) inhibits nuclei neurons during light stimulation. The lower trace is a blown-up view of the boxed area in the upper trace. (**D**) Example of spike firing rate averaged over previous 50 ms at each occurring spike for the 5 s spike train in (**D'**). (**D''**) 1 s spike train for the duration indicated in the square box in (**D'**). Blue bars indicate light stimulation and are specific for ataxia, dystonia, and tremor (see 'Materials and methods' for light stimulation parameters). All example traces originate from the same nuclei neuron, indicating that the cell's spike signature can change depending on the light stimulation paradigm. (**E**) Histograms of instantaneous firing rate (ISI⁻¹) of the full 30 s spike train, observe the shift in distribution from baseline during the different stimulation paradigms. (**F**) Proportion of cells of each predicted spike signature during each of the light stimulations based on classifier model in **Figure 1D**. Control: n = 26 cells, N = 7 mice; ataxia: n = 16, N = 6; dystonia: n = 16, N = 5; tremor n = 16, N = 5. (**G, H**) Scatterplot of spike train parameters used to classify neural signatures. Colored boxes indicate the predicted phenotype, colors of circles indicate the origin of the recorded neurons.

The online version of this article includes the following source data and figure supplement(s) for figure 5:

**Source data 1.** Source data for the graphs in **Figure 5**.

**Figure supplement 1.** Purkinje cells respond differently to specific optogenetic stimulation paradigms.

**Figure supplement 2.** Comparison of group averages per figure.



**Video 2.** Optogenetically induced behavioral responses in head-fixed mice. Subtle features of ataxia, dystonia, and tremor can be induced in head-fixed mice by initiating each specific spike signature. In this experiment, the optopatcher recording and stimulation allowed for tracking the inductions of spike signatures in single units and the subsequent presentation of the disease-associated behaviors. In this video, we show behavioral responses to all stimulation paradigms with their paired recordings of spike activity. All responses and paired spike recordings were from a single recording of the same cell and mouse. Voltage of the spike trace is played as audio.

https://elifesciences.org/articles/91483/figures#video2

cerebellum and that they may be a primary cause for motor impairments associated with cerebellar disease.

It is intriguing that we find multiple cerebellar spike signatures associated with an ataxic phenotype. Our work associates $Pcp2^{Cre};Ank1^{fl/fl}$ and $Atxn1^{154Q/+}$ ataxias with slightly irregular (elevated skewness) and slow or pause-prone (ISI$_{>25}$) firing patterns (*Figure 2*). However, the $Pcp2^{Cre};Slc32a1^{fl/fl}$ ataxia is associated with very regular firing patterns on the scale of both spike-to-spike irregularity (CV2) and that of the entire trace (skewness) (*Figure 1*). In this case, we feel it is essential to consider the underlying pathogenesis of these mouse models (*Supplementary file 1*). The $Pcp2^{Cre};Slc32a1^{fl/fl}$ mouse represents a non-degenerative ataxia with a cell type-specific cerebellar insult (*White et al., 2014*). However, while $Pcp2^{Cre};Ank1^{fl/fl}$ is cerebellar cell type-specific, it is also neurodegenerative (*Stevens et al., 2022*). Meanwhile, $Atxn1^{154Q/+}$ is neither cell type nor cerebellar-specific and is neurodegenerative (*Watase et al., 2002*). Therefore, our work suggests that degenerative ataxias and non-degenerative ataxias may have different underlying circuit alterations that lead to distinct

spike signatures. We also consider the possibility that the complete lack of Purkinje cell GABA neurotransmission in the $Pcp2^{Cre};Slc32a1^{fl/fl}$ mouse allows this high degree of regularity in the cerebellar nuclei cells, while the $Pcp2^{Cre};Ank1^{fl/fl}$ and $Atxn1^{154Q/+}$ mutations likely alter the input from Purkinje cells that the nuclei cells receive, but do not completely remove it. The possibility of any modulation from Purkinje cells likely enables, and perhaps ensures, greater irregularity than that which is possible in the $Pcp2^{Cre};Slc32a1^{fl/fl}$ model. This is somewhat supported by our awake, head-fixed recordings of nuclei cells in response to Purkinje cell terminal channelrhodopsin stimulation, in which we were able to achieve firing patterns more similar to the $Pcp2^{Cre};Ank1^{fl/fl}$ and $Atxn1^{154Q/+}$ signatures than that of $Pcp2^{Cre};Slc32a1^{fl/fl}$ (*Figure 5—figure supplement 2*). Our channelrhodopsin experiment attempts to minimize the modulation of the Purkinje cell input to the cerebellar nuclei, but there is no more minimal modulation of an input than the complete lack of it. However, an important caveat to this interpretation is that our manipulation of the cerebellar nuclei during the awake recordings only impacted Purkinje cell terminals near the nuclei cell being recorded, by design. Therefore, it is possible that with greater synchrony of Purkinje cells (i.e., mimicking the pan-Purkinje cell silencing in the $Pcp2^{Cre};Slc32a1^{fl/fl}$ mouse) a more regular nuclei firing pattern could be induced if the Purkinje cell activity itself has a very regular firing pattern (*Person and Raman, 2012*). Indeed, something closer to this may have been achieved with our bilateral implanted optic fibers during our behavioral experiments. These experiments resulted in distinct behavioral outcomes for each stimulation pattern (*Figure 6*). In short, we have uncovered an intriguing divergence in the spike signatures that are associated with an ataxia-like phenotype. We anticipate that studies into cerebellar neurodegeneration and Purkinje cell synchrony in the context of reduced dynamic range may be fruitful in exploring how these two cerebellar signatures may impact ataxic phenotypes.

Our work suggests that there is a healthy range within the characteristics of cerebellar nuclei spiking activity and that cerebellar movement disorders are associated with a shift from this range in one or multiple features of spike train activity. We find some overlap and shared spike features between the different disease phenotypes and show that healthy cerebellar neurons can adopt multiple disease-associated spike train signatures. These findings suggest that pathophysiological spike train signatures are driven by a shift in neural function toward a disease state that can be independent of plasticity or circuit rewiring. Despite the dramatically different behavioral outcomes, the potential overlap and



**Figure 6.** Induced spike signatures elicit distinct cerebellar phenotypes. (**A**) Schematic of external view of bilateral optical fiber implant. (**B**) Schematic of a coronal section from a mouse cerebellum with bilateral optic fiber implants directed towards the cerebellar nuclei. FN = fastigial nucleus; IN = interposed nucleus; DN = dentate nucleus. (**C**) Photomicrograph of a Nissl-stained coronal section from a mouse cerebellum that had been implanted with optic fibers. Arrows = optic fiber tracks. Dotted lines surround the cerebellar nuclei indicated in (**B**). Scale = 1 cm. (**D**) Schematic of a mouse with bilateral optic fiber implants freely moving in a tremor monitor. (**E–K**) Data associated with gait measurements. (**E–H**) Example footprints from a single mouse before stimulation (baseline, **E**) and during ataxia (**F**), dystonia (**G**), and tremor stimulation (**H**). Scale = 1 cm. (**I–K**) Measurements of gait including the length of the hindpaw stride (**I**), forepaw stride (**J**), and distance between the hind and forepaws (**K**). N = 8 mice. * = p≤0.05; ** = p≤0.01. (**L**) Example

*Figure 6 continued on next page*

*Figure 6 continued*

images of phenotypes associated with dystonia. 1 = erect tail; 2 = high stepping; 3 = kinked tail; 4 = hyperextension of the limbs; 5 = splayed toes. (**M**) Dystonia rating of mice before stimulation and during stimulation with each paradigm. N = 8 mice. * = p≤0.05. (**N**) Tremor signals detected via tremor monitor from a mouse before and during stimulation with each paradigm. Horizontal scale = 1 s. Vertical scale = 50 mV. (**O**) Population average power spectrums of tremor. Solid line = mean power. Shaded region = SEM. (**P**) Peak tremor power of mice before and during stimulation with each paradigm. N = 8 mice. * = p≤0.05; ** = p≤0.01. (**Q–S**) 2-dimensional comparisons of gait (hind to fore distance), dystonia (rating), and tremor (peak power) measurements from all mice. N = 8 mice. (**T**) 3-dimensional plot of data in (**Q–S**).

The online version of this article includes the following source data for figure 6:

**Source data 1.** Source data for the graphs in *Figure 6*.

shared spike features, such as the irregular spike pattern found in the dystonia and tremor signatures, raise the strong possibility that co-expression or comorbidity of different motor disease behaviors may arise due to a spectrum of spike signal defects. This indicates that it is possible for neural signals to shift back and forth between healthy and disease states, potentially resulting in the transientness of behavioral impairments in certain cerebellar disorders. These findings also suggest that the most successful therapeutic avenues for cerebellar movement disorders should maintain cerebellar spiking activity within the healthy range without inadvertently inducing a different pathophysiological signature.

One important question to consider is the identity of the cells that produce these disease-relevant spike signatures. Both Purkinje cells and cerebellar nuclei cells are heterogeneous populations with varied electrophysiological properties (*Zhou et al., 2014*; *Xiao et al., 2014*; *Uusisaari et al., 2007*; *Uusisaari and Knöpfel, 2011*). The recordings we report here were extracellular and performed in awake, head-fixed mice and, therefore, we cannot speak to more precise identities of the cell population included in this work. We can speculate that, as excitatory nuclei cells are larger than inhibitory nuclei cells (*Uusisaari et al., 2007*), our recordings of nuclei cells are more likely to be from excitatory neurons. However, our optogenetic technique uses Purkinje cell terminals to modulate nuclei activity and therefore would impact both excitatory and inhibitory nuclei neurons (*Canto et al., 2016*). Recent work has suggested differential therapeutic benefit and accessibility for treatment for excitatory and inhibitory neurons (*Spix et al., 2021*). Therefore, future studies parsing the impact of excitatory and inhibitory cerebellar neurons on motor impairments would be a great benefit to the field.

There has been (*Slaughter et al., 1970*), and there still remains (*Diniz et al., 2021*), a great interest in understanding how altered cerebellar signals influence human behavior and disease. A pressing need to better define the architecture of these spiking abnormalities has arisen because of the success in using DBS and the potential for better tuning of the stimulation parameters for greater efficacy in treating different cerebellar-related disorders (ataxia: *Teixeira et al., 2015*; *Cury et al., 2022*; dystonia: *Brown et al., 2020a*; tremor: *Horisawa et al., 2021*; *Paraguay et al., 2021*; ataxia, dystonia, tremor: *Diniz et al., 2021*). Recordings in human patients during DBS implantation have previously found physiological differences in the spike train patterns of basal ganglia neurons between dystonia and other movement disorders (e.g., etiologically distinct dystonia: *McClelland et al., 2016*; dystonia with and without head tremor: *Sedov et al., 2020*; dystonia versus Parkinson's disease: *Tang et al.,*

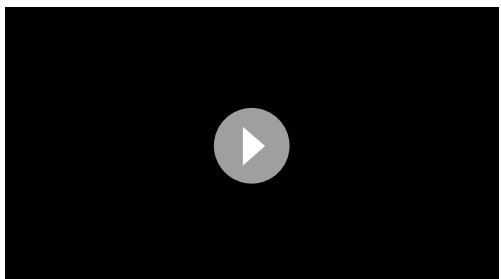

**Video 3.** Optogenetically induced spike signatures result in severe cerebellar phenotypes. Examples of a single freely moving mouse's behavior at baseline and in response to bilateral induction of spike signatures.
https://elifesciences.org/articles/91483/figures#video3

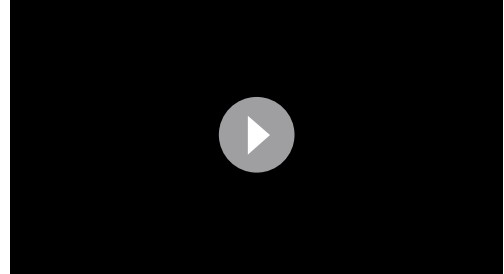

**Video 4.** Optogenetically induced spike signatures affect gait. Examples of a single mouse's gait within a footprinting corridor at baseline and in response to bilateral induction of spike signatures.
https://elifesciences.org/articles/91483/figures#video4

*2007*). These works and ours compared transient, opportunistic recordings of neurons and therefore contain variability imparted from unknown levels of arousal or sensory states. Here we show that these nonsimultaneous recordings from a population of heterogeneous neurons are sufficient to determine distinct pathogenic signatures among the variability of our dataset. However, simultaneous recordings may reveal greater insights into the neural signatures of motor diseases. Simultaneous recording may be beneficial across nodes in the motor circuit (e.g., the basal ganglia and cerebellum in dystonia) or to consider synchrony within a cell population (e.g., across Purkinje cells in ataxia). Indeed, simultaneous recordings can be valuable in determining upstream and downstream nodes in the propagation of pathogenic neural activity, especially if paired with the appearance of disordered movements (*Pedrosa et al., 2014*), which is useful in determining potential therapeutic targets. Our electrophysiological data confirm that differences between disease presentations are found in the spike train properties of the cerebellum and our optogenetic experiments suggest that these spike trains are not merely biomarkers that correlate with the disease state but are sufficient to cause motor impairments. Therefore, it is not clear whether our model would become stronger by considering both cerebellar activity and activity elsewhere in the motor circuit or whether considering multiple areas would be redundant. Our previous work and what we have shown here suggest cerebellar-targeted DBS may be highly beneficial in normalizing disease-causing spike train patterns. As we move toward greater understanding of the motor system's response to such perturbation, it would be beneficial to consider the simultaneous responses of neural activity within and external to the cerebellum to determine ideal biomarkers and therapeutic targets to improve DBS for movement disorders.

A parallel motivation has fueled rodent studies to define the anatomical targets, stimulation parameters, and outcome efficacy in detail (*White and Sillitoe, 2017*; *Miterko et al., 2021*; *Cooperrider et al., 2014*; *Miterko et al., 2019*; *Anderson et al., 2020*). Although the exact mechanism(s) of DBS remains unclear (*Miterko et al., 2019*; *Schor et al., 2022*), there is strong support that at least one target of DBS is the actual neuronal signal itself (which the DBS may modify, enhance, dampen, and perhaps even interfere with the given neural spike defects). Indeed, the ability to alter cerebellar spike activity (*Nashold and Slaughter, 1969*) has driven the investigation of the signal properties (*Slaughter et al., 1970*). Evidently, neurotransmission of the faulty activity patterns is required for the expression of disease behaviors (*Stratton and Lorden, 1991*). Recent works strongly support the hypothesis that changes in activity (rate, pattern, or both) could instigate a range of movement abnormalities (*White and Sillitoe, 2017*; *Brown et al., 2020b*; *Fremont et al., 2014*; *LeDoux and Lorden, 2002*; *Fremont et al., 2017*). However, the current work defines the individual potential of these altered cerebellar signals to initiate specific motor changes. The spike signature associated with neurodegenerative ataxias (*Pcp2^{Cre};Ank1^{fl/fl}* and *Atxn1^{154Q/+}*) has slightly elevated skewness and slower or pause-prone firing compared to control (ISI$_{>25}$, *Figure 5—figure supplement 2*) while exceedingly steady activity is associated with the nondegenerative *Pcp2^{Cre};Slc32a1^{fl/fl}* ataxia. Modulation of interposed nucleus activity alters movement trajectories (*Becker and Person, 2019*). Therefore, incorrect modulation of nuclei activity in the neurodegenerative ataxias or lack of correct modulation in *Pcp2^{Cre};Slc32a1^{fl/fl}* ataxia may lead to the ataxia phenotype. The dystonic models herein were characterized with irregular and slow firing. Previous work has associated bursts of Purkinje cell activity with dystonic postures (*Hisatsune et al., 2013*), and therefore, the suppressed periods of the irregular pattern associated with dystonia may be related to the timing of dystonic muscle contractions. Indeed, in our previous work we have shown that stimulation of Purkinje cell terminals in the cerebellar nuclei, resulting in suppression of cerebellar nuclei activity, directly precedes muscle contractions (*Brown et al., 2020b*). Finally, a rhythmic burst pattern of activity was associated with a tremor signature. A large body of work implicates oscillatory activity propagating through the motor network to induce rhythmic oscillations of muscle activity in tremor (*Brown et al., 2020b*; *Pedrosa et al., 2014*; *Hua et al., 1998*). Indeed, bursting activity in the cerebellar nuclei has a rather direct relationship to the resultant tremor (*Brown et al., 2020b*). For all phenotypes, we expect that a large proportion of the population of neurons must produce the abnormal spike signature in order to produce a severe motor phenotype (*Figures 2 and 3*). Our optogenetic behavioral studies likely induced synchrony of both Purkinje and nuclei cell activity in addition to the intended spike signature; therefore, our work here supports synchronous abnormal firing for the phenotypes we produced. However, we cannot claim that synchrony is necessary to produce these phenotypes. Together, these data unveil a critical role for the cerebellum in triggering disease behaviors, with causative signals likely originating locally

in its circuitry. Each disease may arise as a result of a change in the balance and representation of neural signatures.

# Materials and methods

## Key resources table

| Reagent type (species) or resource | Designation | Source or reference | Identifiers | Additional information |
|---|---|---|---|---|
| Strain, strain background (*Mus musculus*, female and male) | *Pcp2$^{Cre}$* | *Lewis et al., 2004*; DOI:10.1016/j.ydbio.2004.03.007 | | |
| Strain, strain background (*M. musculus*, female and male) | *Ptf1a$^{Cre}$* | *Kawaguchi et al., 2002*; DOI:10.1038/ng959 | | |
| Strain, strain background (*M. musculus*, female and male) | *Slc32a1$^{fl}$* | JAX | #012897 | |
| Strain, strain background (*M. musculus*, female and male) | *Slc17a6$^{fl}$* | JAX | #012898 | |
| Strain, strain background (*M. musculus*, female and male) | *Pdx1$^{Cre}$* | *Gu et al., 2002*; DOI:10.1242/dev.129.10.2447 | | |
| Strain, strain background (*M. musculus*, female and male) | *Car8$^{wdl/wdl}$* | JAX | #004625 | |
| Strain, strain background (*M. musculus*, female and male) | *Atxn1$^{154Q}$* | JAX | #005601 | |
| Strain, strain background (*M. musculus*, female and male) | *Ank1$^{fl}$* | JAX | #036512 | |
| Strain, strain background (*M. musculus*, female and male) | *Rosa26$^{lsl-ChR2-eYFP}$* | JAX | #024109 | |
| Chemical compound, drug | Harmaline | Sigma-Aldrich | #H1392 | 30 mg/kg |
| Chemical compound, drug | 2,2,2-Tribromoethanol | Sigma-Aldrich | #T48402 | |
| Software, algorithm | Spike2 | CED | RRID:SCR_000903 | |
| Software, algorithm | MATLAB | MathWorks | RRID:SCR_001622 | |
| Software, algorithm | GraphPad Prism | GraphPad Software | RRID:SCR_002798 | |
| Software, algorithm | Photoshop | Adobe | RRID:SCR_014199 | |
| Software, algorithm | Illustrator | Adobe | RRID:SCR_010279 | |
| Software, algorithm | Leica Application Suite X (LAS X) | Leica Microsystems | RRID:SCR_013673 | |
| Other | Tissue-Tek O.C.T. Compound | VWR | #25608-930 | Specimen embedding compound for cryostat sectioning |
| Other | Cresyl violet acetate 0.1% aqueous | Electron Microscopy Sciences | #26089-01 | Tissue staining solution |
| Other | Cytoseal XYL | Electron Microscopy Sciences | #18009 | Mounting media |

## Animals

Mice were housed in a level 3, AALAS-certified vivarium. The Institutional Animal Care and Use Committee of Baylor College of Medicine reviewed and approved all experimental procedures that involved mice. We used the following transgenic mouse lines: *Pcp2$^{Cre}$* (also known as *L7$^{Cre}$*) (*Lewis et al., 2004*), *Slc32a1$^{fl}$* (also known as *Vgat$^{fl}$*; JAX:012897) (*Tong et al., 2008*), *Ptf1a$^{Cre}$* (gift from Dr. Chris Wright) (*Kawaguchi et al., 2002*), *Slc17a6$^{fl}$* (also known as *Vglut2$^{fl}$*; JAX:012898) (*Tong et al., 2007*), *Pdx1$^{Cre}$* (gift from Dr. Qingchun Tong) (*Gu et al., 2002*), *Car8$^{wdl/wdl}$* (JAX:004625) (*Jiao et al., 2005*), *Atxn1$^{154Q}$* (JAX:005601) (*Watase et al., 2002*), *Ank1$^{fl}$* (JAX:036512) (*Stevens et al., 2021*), and *Ai32* (*Rosa26$^{lsl-ChR2-eYFP}$*, JAX:024109) (*Madisen et al., 2012*). We included mice from both sexes. We used ear punches from pre-weaned pups for PCR genotyping to identify the different transgenic alleles. Mating strategies ensured only heterozygous expression of *Cre*.

## Headplate and craniotomy surgery for electrophysiology recordings

Prior to all recordings, we performed a surgery to attach a headplate over Bregma and make a craniotomy over the interposed nucleus. This allowed us to collect stable recordings of cerebellar neuron activity while the mouse was awake and with or without a severe motor phenotype. This surgery was detailed in our previous publication (*White et al., 2016b*). In short, mice were given preemptive analgesics, including a subcutaneous injection of buprenorphine and meloxicam. Anesthesia was induced using inhaled isoflurane. During the surgery, the mice were kept on a heating blanket to maintain body temperature. Fur was removed from the surgery site using depilatory cream (Nair) and a small incision in the skin was made over the skull. Next, we used a dental drill to make an ~2-mm-diameter craniotomy over the interposed nucleus (6.4 mm posterior and 1.3 mm lateral to Bregma). The craniotomy was surrounded by a round chamber and filled with antibiotic ointment until the day of the recording. The recording chamber was closed with a screw top or silicone cap. We also placed a headplate with a hole over Bregma. A small piece of wire was placed vertically over Bregma to identify this anatomical marker on the day of recording. The recording chamber and headplate were attached to the skull using a combination of C and B Metabond Adhesive Luting Cement (Parkell) and methyl methacrylate dental cement (A-M Systems). Mice were allowed to recover from the surgery for at least 3 days before we began conducting the in vivo electrophysiological recordings.

## Standard in vivo electrophysiological recordings in awake mice

We performed in vivo electrophysiology recordings according to experimental procedures detailed in our previous publications (*Brown et al., 2020b*; *White et al., 2016b*; *van der Heijden et al., 2021a*). In brief, we placed mice on a rotating foam wheel and stabilized their heads by screwing the implanted headplates to the recording rig. We removed the antibiotic ointment from the recording chamber and replaced it with sterile physiological saline solution (unless otherwise specified). We measured the coordinates of Bregma to determine the coordinates where the electrode would penetrate the surface of the cerebellum. We used tungsten electrodes (2–8 MΩ, Thomas Recording) attached to a preamplifier head stage (NPI Electronic instruments) for our recordings. The position of the recording electrode was controlled using a motorized micromanipulator (MP-255; Sutter Instrument Co). Neural signals were amplified and bandpass filtered (0.3–13 kHz) using an ELC-03XS amplifier (NPI Electronics) before being digitized using a CED board. All signals were recorded using Spike2 software (CED). We only included neurons recorded between 2.5 and 3.5 mm from the surface that did not exhibit complex spikes (characteristic for Purkinje cell firing) in our analyses of cerebellar nuclei neuron firing patterns. This article includes reanalyzed data from previously reported studies (*White and Sillitoe, 2017*; *Brown et al., 2020b*; *Stevens et al., 2022*; *van der Heijden et al., 2021a*).

## Spike sorting and analyses

We previously showed that recording duration influences the estimation of neural firing parameters (*van der Heijden et al., 2022b*). Therefore, all parameter estimates in this article are based on 30-s-long recordings. We used Spike2 software to sort out spikes from electrophysiological recordings. Information about spike timing was imported and saved in MATLAB databases using custom written code. We described the spike analyses using 12 parameters investigating the properties of ISIs within the recording:

$$firing\ rate = \frac{spikes}{s}; \tag{1}$$

$$mean\ instantaneous\ firing\ rate = mean\left(ISI^{-1}\right); \tag{2}$$

$$median\ instantaneous\ firing\ rate = median\left(ISI^{-1}\right); \tag{3}$$

$$CV = \frac{stdev\left(ISI\right)}{mean\left(ISI\right)}; \tag{4}$$

$$CV2 = mean\left(2 \times \left|\frac{ISI_n - ISI_{n-1}}{ISI_n + ISI_{n-1}}\right|\right); \tag{5}$$

$$skewness = 2 * \frac{median\ instantaneous\ firing\ rate\ -\ firing\ rate}{median\ instantaneous\ firing\ rate\ +\ firing\ rate}; \tag{6}$$

$$ISI_{25} = \% \, ISI > 25 \, ms; \tag{7}$$

$$ISI_{100} = \% \, ISI > 100 \, ms; \tag{8}$$

$$inter \, burst \, pause = mean\left(ISI > \left(5 * mean\left(ISI\right)\right)\right); \tag{9}$$

$$kurtosis = \% \, ISI \, at \, mode\left(ISI\right); \tag{10}$$

$$rhythmicity \, index = \frac{a_1}{z} + \frac{b_1}{z} + \frac{a_2}{z} + \frac{b_2}{z} + \ldots; \tag{11}$$

$$oscillation \, peaks = number \, of \, peaks \, "a". \tag{12}$$

For the calculation of *Equation 11* and *Equation 12*, we performed an autocorrelation analysis for all spikes in the 30 s recording, calculated the rhythmicity index, and found oscillation peaks as previously described (*Lang et al., 1997*; *van der Heijden et al., 2021b*). We used a bin width of 5 ms. The first oscillation peak was determined as the highest bin between 10 ms and 1.5 times the mean ISI for a given neuron. The difference between the baseline level and the height of the peak is denoted as $a_1$, the difference between baseline and the depth of the trough is denoted as $b_1$, and $z$ is the difference between baseline and the total number of spikes included in the analyses. Each subsequent peak was determined as the highest bin between the delay time of the previous trough and $a_n + a_1 + 10$ ms, where $a_n$ is the time of the previous peak. The first trough was determined as the lowest bin between the first peak ($a_1$) and $a_n + a_1$. Peaks and troughs were only accepted if their sum was higher than four times the standard deviation of autocorrelation between 0.96 and 1 s lag-time or if the peak was higher than baseline plus two times the standard deviation and the trough was lower than baseline minus two times the standard deviation.

## Supervised classifier learner

We trained our classifier learner using cerebellar nuclei neuron recordings from mixed background control mice (control), *Pcp2Cre;Slc32a1fl/fl* mice (ataxia), *Ptf1aCre;Slc17a6fl/fl* mice (dystonia), and harmaline-injected mice (tremor). We have previously tested these models and consider them as reliable archetypical models for cerebellar movement disorders (*White et al., 2014*; *White and Sillitoe, 2017*; *Brown et al., 2020b*). For the analyses, we reanalyzed previous recordings (*White and Sillitoe, 2017*; *Brown et al., 2020b*; *Stevens et al., 2022*; *Zhou et al., 2022*; *van der Heijden et al., 2021a*) in addition to newly acquired ones. We previously found that the recording duration used for parameter analyses influences the presentation of spike properties. Therefore, we only included recordings with a duration of exactly 30 s and calculated parameter estimates as described above (*van der Heijden et al., 2022b*). Our previous study also showed that cerebellar nuclei neurons recorded in the same mouse had equal parameter variability as nuclei neurons recorded in different mice. Therefore, we treated all single-nuclei neuron recordings as independent datapoints (*van der Heijden et al., 2022b*).

We used the built-in supervised machine learning, MATLAB Classification Learner APP (The Math-Works Inc, version R2021a) to define spike signature parameter value cutoffs. We imported the parameters describing the spiking activity from an Excel (Microsoft) worksheet. We used 'Group' (control, ataxia, dystonia, or tremor) as the 'Response Variable' and the 12 parameters described above as the 'Predictor Variables'. We trained a 'Coarse Tree' with maximum number of splits = 3, split criterion = Gini's diversity index, and surrogate decision splits = off. We exported the trained model to the workspace to validate the classifier learner. We chose to use a coarse classifier learner to find unique spike signatures because spike parameters are highly correlated with each other (*Figure 1—figure supplement 2*), and we wanted to prevent overfitting of our model to correlated parameters based on a relatively limited set of recordings.

We trained 12 classifier models based on training sets of control (n = 25), ataxia (n = 20), dystonia (n = 20), and tremor (n = 20) neurons, and 12 distinct validation sets of control (n = 8), ataxia (n = 4), dystonia (n = 4), and tremor (n = 4) neurons. We validated how the classifier models (see *Figure 1—figure supplement 3*) performed and used the classifier with the best performance throughout the article to assign signatures to nuclei cell spike patterns.

## Optopatcher in vivo electrophysiology recordings in awake mice

Optopatcher experiments were performed as previously described (*van der Heijden et al., 2022a*). We prepared the mice for recording by performing a headplate and craniotomy surgery as described above. The electrophysiology rig and setup were the same as described above with the following differences: we used an optopatcher device (ALA Scientific Instruments Inc) that allows for the placement of a custom-made optical fiber (ThorLabs, #FT200UMT) within a glass recording electrode (Harvard Apparatus, #30-0057). The tip of the optical fiber was placed near the tip of the recording electrode and was illuminated via a 465 nm LED light source (ALA Scientific Instruments Inc). On the day of the recording, we pulled glass electrodes (Sutter Instruments, #P-1000), filled the electrodes with physiological saline, and measured their impedance using an ELC-03XS amplifier (NPI Electronics) before recording. Only electrodes with 2–15 MΩ impedance were used. Light stimulation was triggered using custom Spike2 scripts paired with a CED board (CED). All optopatcher recordings were performed in *Pcp2^{Cre};Ai32* mice that express channelrhodopsin in Purkinje cells. Nuclei neurons included in our analyses were cell recordings between 2.5–3.5 mm from the surface of the cerebellum and were inhibited by brief light stimulation at 465 nm. After we found and isolated a cell, we slowly ramped up the brightness of this brief stimulation until we found the minimal light power that modulated the spiking activity. This minimal power was then used to stimulate the cell with the various test parameters during the recording session. Only cells with a 'control' spike signature at baseline were considered for further analysis.

## Optogenetic light stimulation paradigms for optopatcher recordings and behavioral assays

We used the following stimulation parameters to measure signature-specific responses; ataxia: 50 Hz (10 ms light on/10 ms light off) square pulses; dystonia: at least 1 s of 50 Hz (10 ms light on/10 ms light off) square pulses interspersed with at 75% chance of 250 ms pauses in stimulation; tremor: 100 Hz pulses (5 ms light on/5 ms light off) overlaid with a 10 Hz sinusoid (50 ms parabolic increase and decrease of light power followed by 50 ms light off).

## Bilateral optic fiber implant surgery for in vivo behavioral assays

We implanted mice with optical fibers (ThorLabs, #FT200UMT) targeted bilaterally to the interposed cerebellar nuclei to assess the motor phenotypes that result from our stimulation parameters. We previously described this surgical procedure (*Brown et al., 2020b*). Briefly, the mice were given subcutaneous injections of sustained-release buprenorphine and meloxicam as preemptive analgesics. Anesthesia was induced with 3% isoflurane gas and the anesthetic plane was maintained with 2% isoflurane gas. The mice were placed on a heating blanket and the head was stabilized in a stereotaxic frame (David Kopf Instruments). Fur was removed from the top of the head using depilatory cream (Nair). The surgical site was cleaned using alternating applications of betadine scrub and alcohol. An incision was made in the skin to expose the skull from anterior to Bregma to the posterior aspect of the occipital plate. The fascia was removed from the top of the skull and a scalpel was used to etch fine grooves into the top of the skull. A small craniotomy was made in the parietal plate with a dental drill in order to attach a skull screw (00–90 × 1/16 flat point stainless steel machine screw) to anchor the implant to the skull. The skull screw was advanced only until the point that it was stable in the skull and care was taken to ensure it did not contact the brain. The implant sites were determined by measuring 6.4 mm posterior and ±1.5 mm lateral to Bregma. Small craniotomies were made at these sites using a dental drill. The base of the fibers were placed on the surface of the cerebellum and was advanced ventrally for 2 mm. A small amount of antibiotic ointment was placed around each fiber to prevent Metabond or dental cement from entering the craniotomy. Metabond was applied over the entire exposed area of skull and around the skull screw and fibers. Two short segments of wire were embedded in the Metabond to allow the experimenter to hold the mouse's head while attaching and removing the fiber patch cables from the implant. Dental cement was placed over the Metabond. The mice were allowed to recover for at least 3 days before any behavioral assessments were made. Subcutaneous injections of meloxicam were provided every 24 hr during the recovery period. Eight mice (number of mice, N) were implanted and tested with all three behavioral assays.

## Gait analysis

Measurements of gait were made for all mice both before (baseline) and during each stimulation parameter. To do this, the mice were briefly scruffed and then contrasting colored non-toxic paints were applied to the soles of their forepaws and hindpaws (blue paint for forepaws and red paint for hindpaws). The mice were then gently set down on a piece of blank white paper positioned between two parallel plexiglass barriers with a dark enclosed area at the end of the corridor. Adult mice are naturally inclined to walk toward the dark enclosed area, meanwhile depositing paint on the paper with each footstep. The stimulation, if present, was extinguished once the mice entered the enclosed area. The mice were allowed to rest for at least 2 min between each trial. A trial was considered successful if there were at least four hind and forepaw prints for both the left and right feet that were visible, steady (the mouse was neither running or stopping midway through the series of foot-steps), and in as straight of a line as possible (the mouse was not actively turning during the trial). Optic patch cables were connected to the implant during every trial (including baseline runs when stimulation was not present). At least two successful trails were collected per stimulation parameter. For analysis, measurements from two trials of the same type were averaged to determine the values for each mouse. These measurements were stride, the distance from one footprint to the next from the same foot and hind-to-fore distance, the distance from one hindpaw placement from a corre-sponding forepaw placement on the same side of the body. Three of each measurement were made per footprint and were averaged to determine the measurements per foot, per trial. These trials were then averaged to determine the final measurements for each mouse. A repeated-measures one-way ANOVA with a Tukey multiple comparison adjustment was performed using GraphPad Prism software (GraphPad Software, La Jolla, CA) to determine significance. Significance values are indicated as not significant (ns) if $p>0.05$, * = $p\leq0.05$, ** = $p\leq0.01$.

## Dystonia rating scale

Mice were placed in a rectangular plexiglass arena in order to phenotypically rate the frequency and severity of dystonic behavior as described previously (*Jinnah et al., 2000*; *Pizoli et al., 2002*). The videos were captured to include the animals' behavior before (baseline) and during each stimulation paradigm (ataxia, dystonia, and tremor). Each stimulation period lasted 2 min, during which the mouse was allowed to ambulate freely as well as in response to disturbance by the experimenter. The rating period excluded the first 10 s after stimulation was initiated to avoid including behavior in response to the sudden application of stimulation. Mice were given a score of 0 if no motor abnormalities were identified, 1 if there was abnormal motor behavior that was not obviously dystonic (such as severe tremor), 2 if there was a mild motor phenotype that included dystonic behaviors that may occur only in response to being disturbed by the experimenter or unprovoked (such as brief hyperextension of the limbs or extension of the neck and/or back), 3 if there was moderate severity or frequent unprovoked dystonic behaviors, 4 if there were severe unprovoked dystonic behaviors, and 5 if there were severe unprovoked dystonic behaviors that made goal-directed ambulation extremely difficult or impossible for an extended time. A detailed description of dystonic phenotypes in mice can be found in *Brown et al., 2023*. Wilcoxon matched-pairs signed rank tests were performed with a post hoc Holm–Sidak method for p-value adjustment in order to determine significance using GraphPad Prism software (GraphPad Software). Significance values are indicated as not significant (ns) if $p>0.05$, * = $p\leq0.05$, ** = $p\leq0.01$.

## Tremor analysis

The amplitude and frequency of tremor before (baseline) and during optogenetic stimulation were measured using a custom-made tremor monitor as previously described (*Brown et al., 2020b*; *Zhou et al., 2022*; *van der Heijden et al., 2021a*). The mice were implanted with optical fibers targeted to the interposed nucleus as described above. LED drivers were connected to the implant via optical patch cables and placed above the tremor monitor chamber. The tremor monitor chamber is a clear plexiglass box that is suspended by elastic cords that are connected to each corner of the box. An accelerometer is securely mounted to the bottom of the box to detect the tremor of the mouse within the box. The mice were placed in the chamber and were able to freely ambulate while in the box. They were given 2 min to acclimate to the novel tremor monitor environment before 2 min duration baseline recordings were made. The mice were then stimulated with the ataxia, dystonia,

and tremor stimulation parameters for 2 min per stimulation period. At least 2 min were allowed to elapse between stimulation periods. Mice were encouraged to ambulate a similar amount during all recording periods. The output from the tremor monitor was amplified and lowpass filtered at 5 kHz (Brownlee Precision, Model 410) before being digitized by a CED board and analyzed using Spike2 scripts. Tremor monitor recordings were centered on zero and downsampled using the Spike2 interpolate function. A power spectrum analysis with a Hanning window was performed on each recording period. The peak power was calculated as the maximum power between 0 and 30 Hz, which is consistent with the range that we typically observe for physiological and pathological tremor in mice (*Brown et al., 2020b*). A repeated-measures one-way ANOVA with Tukey multiple comparison adjustment was performed using GraphPad Prism software (GraphPad Software) to determine significance. Significance values are indicated as not significant (ns) if $p>0.05$, * = $p≤0.05$, ** = $p≤0.01$.

### Histology

After electrophysiology and behavior experimentation, mice were heavily anesthetized with avertin (2,2,2-tribromoethanol, Sigma-Aldrich, St. Louis, MO; #T48402) and transcardially perfused with ice-cold phosphate buffered saline (PBS, 1×) followed by ice-cold 4% paraformaldehyde (PFA). The implants were removed, and then the brains were dissected out of the skull. The brains were post-fixed in PFA at 4°C for at least 24 hr. They were then cryoprotected via a sucrose gradient, starting at 15% sucrose in PBS followed by 30% sucrose in PBS. The brains were imbedded in Tissue-Tek O.C.T. Compound (Sakura, Torrance, CA), frozen, and then sliced on a cryostat to produce 40 µm thickness sections. The tissue sections were washed in PBS, mounted onto slides, and allowed to dry on the slide for at least 2 hr. The sections were put in a xylene and ethanol series with ~2 min per submersion in the following order: 100% xylene, 100% xylene, 100% ethanol, 100% ethanol, 90% ethanol, 70% ethanol. The sections were then placed in water for ~2 min and then submerged in cresyl violet solution until the stain was sufficiently dark. They were then dehydrated by reversing the series with ~30 s per submersion. Finally, Cytoseal XYL mounting media (Thermo Scientific, Waltham, MA, #22-050-262) and a cover slip were placed on the slides. The slides were allowed to dry in a fume hood before imaging. Photomicrographs were obtained using a Leica DM4000 B LED microscope equipped with a Leica DMC 2900 camera and Leica Application Suite X (LAS X) software. Images were corrected for brightness and contrast using Adobe Photoshop (Adobe Systems, San Jose, CA). Figure panels were made using Adobe Illustrator software.

### Acknowledgements

This work was supported by the Baylor College of Medicine (BCM), Texas Children's Hospital, The Hamill Foundation, and the National Institutes of Neurological Disorders and Stroke (NINDS) R01NS100874, R01NS119301, and R01NS127435 to RVS. Research reported in this publication was supported by the Eunice Kennedy Shriver National Institute of Child Health & Human Development of the National Institutes of Health under award number P50HD103555 for use of the Cell and Tissue Pathogenesis Core and In Situ Hybridization Core (the BCM IDDRC). This work was also supported by the BCM Bioengineering Core with the expert assistance of I-Chih Tan. The content is solely the responsibility of the authors and does not necessarily represent the official views of the National Institutes of Health. MEvdH was supported by a postdoctoral award from the Dystonia Medical Research Foundation (DMRF), the K99NS130463, and start-up funds provided by the Red Gates Foundation and Virginia Polytechnic Institute and State University. A portion of the data was collected when AMB was supported by F31NS101891.

### Additional information

#### Competing interests
Roy V Sillitoe: Reviewing editor, *eLife*. The other authors declare that no competing interests exist.

## Funding

| Funder | Grant reference number | Author |
| --- | --- | --- |
| National Institute of Neurological Disorders and Stroke | R01NS100874 | Roy V Sillitoe |
| National Institute of Neurological Disorders and Stroke | R01NS119301 | Roy V Sillitoe |
| National Institute of Neurological Disorders and Stroke | R01NS127435 | Roy V Sillitoe |
| Eunice Kennedy Shriver National Institute of Child Health and Human Development | P50HD103555 | Roy V Sillitoe |
| Dystonia Medical Research Foundation | | Meike E van der Heijden |
| National Institute of Neurological Disorders and Stroke | K99NS130463 | Meike E van der Heijden |
| Virginia Polytechnic Institute and State University | | Meike E van der Heijden |
| Red Gates Foundation | | Meike E van der Heijden |
| National Institute of Neurological Disorders and Stroke | F31NS101891 | Amanda M Brown |
| Baylor College of Medicine | | Roy V Sillitoe |
| Texas Children's Hospital | | Roy V Sillitoe |
| Hamill Foundation | | Roy V Sillitoe |

The funders had no role in study design, data collection and interpretation, or the decision to submit the work for publication.

## Author contributions

Meike E van der Heijden, Amanda M Brown, Conceptualization, Data curation, Software, Formal analysis, Funding acquisition, Validation, Investigation, Visualization, Methodology, Writing – original draft, Project administration, Writing – review and editing; Dominic J Kizek, Data curation, Investigation; Roy V Sillitoe, Conceptualization, Resources, Formal analysis, Supervision, Funding acquisition, Investigation, Methodology, Writing – original draft, Writing – review and editing

## Author ORCIDs

Meike E van der Heijden ⓘ https://orcid.org/0000-0003-0801-8806
Amanda M Brown ⓘ https://orcid.org/0000-0002-1484-8972
Roy V Sillitoe ⓘ https://orcid.org/0000-0002-6177-6190

## Ethics

Mice were housed in a Level 3, AALAS-certified vivarium. The Institutional Animal Care and Use Committee (IACUC) of Baylor College of Medicine (BCM) reviewed and approved all experimental procedures that involved mice (animal protocol number AN-5996).

Reviewer #1 (Public Review): https://doi.org/10.7554/eLife.91483.3.sa1
Reviewer #2 (Public Review): https://doi.org/10.7554/eLife.91483.3.sa2
Reviewer #3 (Public Review): https://doi.org/10.7554/eLife.91483.3.sa3
Author response https://doi.org/10.7554/eLife.91483.3.sa4

# Additional files

## Supplementary files

• Supplementary file 1. Validation of mouse models of motor disorders. Each model used in this article is listed in the table with the type of model, that is, the predominant phenotype reported for the model, motor behaviors that have been quantified relative to control animals (effect direction is noted when difference is statistically significant), and anatomical changes associated with the model. Only congruent and/or undisputed findings in mice are included. Abbreviations used: Purkinje cell (PC), vesicular GABA transporter (Slc32a1), immunohistochemistry (IHC), wheat germ agglutinin (WGA), tyrosine hydroxylase (TH), vesicular glutamate transporter 2 (Slc17a6), inferior olive (IO), electromyography (EMG), cerebellar nuclei (CN), spinocerebellar ataxia type 1 (SCA1), carbonic anhydrase-related protein 8 (Car8), granule cell (GC), climbing fiber (CF), mossy fiber (MF), lateral hypothalamus (LH), and not available (NA).

• MDAR checklist

## Data availability

All data is available in the main text, supplementary materials, or supporting files.

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
