## [Editor Report · eLife assessment]

The authors' dataset and analysis provide a **fundamental** new understanding of how cerebellar output contributes to various cerebellar-dependent diseases. The observation that different firing statistics at the level of the cerebellar nuclei directly impart disease-specific phenotypes is quite **convincing**. The classifier used in the article remains a potential weak point, showing limited efficacy, particularly for identifying mice with tremor. The concern about classifier accuracy is ameliorated by the fact that the classifier parameters are easily interpretable, and allowed the authors to use these parameters to design stimulation experiments.

---

## [Referee Report · Reviewer #1 (Public Review)]

Summary:

van der Heijden et al perform an ambitious analysis of single unit activity in the interposed nuclei of multiple mouse models of cerebellar dysfunction. Based on these recordings, they develop a classifier to predict the behavioral phenotype (ataxic, dystonic, or tremor) of each model, suggesting that highly regular spiking is associated with ataxia, irregular spiking is associated with dystonia, and rhythmic spiking is associated with tremor. Interestingly, the "dystonic" and "tremor" patterns appeared to be specific to those disorders, while ataxia could result from at least two different interposed nucleus firing patterns. After developing this classifier, they show that activating Purkinje neurons in different patterns that evoke interposed nuclear activity similar to their "ataxic", "dystonic", and "tremor" firing patterns induce similar behaviors in healthy mice. These results show convincingly that specific patterns of cerebellar output are sufficient to cause specific movement abnormalities. The extent to which cerebellar nuclear firing patterns are solely responsible for phenotypes in human disease remains to be established, however.

Strengths:

Major strengths are the recordings across multiple phenotypic models including genetic and pharmacologic manipulations, and the robust phenotypes elicited by Purkinje neuron stimulation.

Weaknesses:

The number of units recorded was small for each model (on the order of 20), limiting conclusions that can be drawn from the recording/classifier experiments.

---

## [Referee Report · Reviewer #2 (Public Review)]

Cerebellar diseases can manifest as various behavioral phenotypes, such as ataxia, dystonia, and tremor. In this study, van der Heijden and colleagues aim to understand whether these differing behavioral phenotypes are associated with disease-specific changes in the firing patterns of cerebellar output neurons in the cerebellar nuclei (CN). The authors effectively demonstrate that across different mouse models of cerebellar disease, there are distinct changes in the firing properties of CN neurons. They take a crucial step further by attempting to replicate disease-specific firing patterns in the cerebellar output neurons of healthy (control) mice using optogenetics. When Purkinje cells are stimulated in a manner that results in similar firing properties in CN neurons, the authors observe a variety of atypical behavioral responses, many of which align with the behavioral phenotypes observed in mouse models of the respective diseases.

Overall, the primary results are quite convincing. Specifically, they show that (1) different mouse models of cerebellar disease exhibit different statistics of firing in CN neurons, and (2) driving CN neurons in a time-varying manner that mimics the statistics measured in disease models results in behavioral phenomena reminiscent of the disease states. These findings suggest that aberrant activity in the CN can originate from various sources (e.g., developmental circuit deficits, abnormal plasticity, insult), but ultimately, these changes are funneled through the CN neurons, whose firing rates are affected, and this, in turn, drives some portion of the aberrant behavior. This is a noteworthy observation that underscores the potential of targeting these output neurons in the treatment of cerebellar disease. Moreover, this manuscript provides valuable insights into the firing patterns associated with the most common cerebellar-dependent disease phenotypes.

However, the applicability of the classifier for identifying mice cerebellar behavioral phenotypes directly from the spiking activity of neurons in the cerebellar nuclei remains this paper's weak point. Cross-validated performance of the model on a single mouse model of tremor is, for instance, only 54%. However, a benefit of this classifier is its overall simplicity; only three parameters are required to achieve average classifier performance of 76%. While more sophisticated models might provide improved classifier performance and enhanced generalization, such models would suffer from a lack of interpretability. This paper, therefore, represents a reasonable starting point for understanding the parameter space of cerebellar nuclei firing and its relationship to behavioral phenotypes during disease.

---

## [Referee Report · Reviewer #3 (Public Review)]

Summary:

This manuscript looks at the single-cell spike signatures taken from in vivo cerebellar nuclear neurons from awake mice suffering from 3 distinct diseases and uses a sophisticated classifier model to predict disease based on a number of different parameters about the spiking patterns, rather than just one or two. Single read-outs of spike firing patterns did not show significant differences between all 4 groups meaning that you need to analyze multiple parameters of the spike trains to get this information. The results are really satisfying and intriguing, with some diseases separating very well, and others having more overlap. It also represents a significant advancement for the rigor and creativity used for analyzing cerebellar output spike patterns. I really like this paper, it's a clever idea and has been done very well.

The authors examine multiple distinct forms of different diseases, including different types of ataxia, dystonia, and tremor. While some of the interpretation of this work remains unclear to this reviewer (in particular Fig. 2, with ataxia models), I applaud the rigor, and sharing complex data that is not always straightforward to understand.

Strengths:

The work is technically impressive and the analysis pushes the envelope of how cerebellar dysfunction is classified, which makes it an important paper for the field.

It's well written. The approach it is taking is clever. The analysis is thorough, and the authors examine a wide array of different disease models, which is time-consuming, costly, and very challenging to do. It's a very strong manuscript.

---

## [Author Response]

The following is the authors’ response to the current reviews.

**Reviewer #1 (Recommendations For The Authors):**
I still find it really impressive that the Purkinje cell stimulation so closely mimics the pathogenic phenotypes - in my opinion, the strongest part of the paper. I would like just a little clarification on some of my previous questions.Major points:(1) Can the authors clarify where the new units came from? Are these units that were recorded before the initial submission and excluded, but are now included? If so, why were they excluded before? Or are these units that were recorded since the original submission?

The number of units increased in Figure 1 for three reasons: (1) We have now plotted the classifier results in Figure 1 instead of the validation results, which have been moved to Figure 1 Supplement 3. (2) In response to reviewer comments, we no longer include units that had >60 s of recording in both our model creation and validation. We had previously used 30 s for creating the model and a different 30 s for validating the model, if an additional 30 s were available. (3) We changed our model creation and validation strategy based on previous reviewer comments. The new units in Figures 2-4 were taken from our pool of previously collected but unanalyzed data (we collect neural data on a rolling basis and thus these data were not initially available). We were fortunate to have these data to analyze in order to address the concerns about the number of cells included in the manuscript. The number of units increased in Figure 5 because new units were recorded in response to reviewer comments.

(2) Why did some of the neuron counts go down? For example, in Pdx1Cre;Vglut2fl/fl mice, the fraction of units with the control signature went from 11/21 to 7/23. Is this because the classifier changed between the original submission and the revision?

Yes, the proportion of cells matching each classification changed due to the different parameters and thresholds used in the updated classifier model.

Minor points:In the Discussion: "We find some overlap and shared spike features between the different disease phenotypes and show that healthy cerebellar neurons can adapt multiple disease-associated spike train signatures." I think "adapt" should be "adopt"In the Discussion: "compare" is misspelled as "compared"

Thank you for bringing these typos to our attention. We will upload a new version of the text with the typos corrected.

The following is the authors’ response to the original reviews.

We would like to thank the Reviewers for providing excellent and constructive suggestions that have enabled us to strengthen our overall presentation of our data. We have addressed each of the comments by altering the text, providing additional data, and revising the figures, as requested.

Below are our explanations for how we have altered the manuscript in this revised version.

**Recommendations for the authors:**
I think you will have seen from the comments that there was great enthusiasm for the importance of this study. There were also shared concerns about how the classifier may be inadequate in its current format, as well as specific suggestions to consider to improve. I hope that you will consider a revision to really amplify the impact of the importance of this study.
**Reviewer #1 (Recommendations For The Authors):**
Distinct motor phenotypes are reflected in different neuronal firing patterns at different loci in motor circuits. However, it is difficult to determine if these altered firing patterns: (1) reflect the underlying neuropathology or phenotype, (2) whether these changes are intrinsic to the local cell population or caused by larger network changes, and (3) whether abnormal firing patterns cause or reflect abnormal movement patterns. This manuscript attempts to address these questions by recording neural firing patterns in deep cerebellar nucleus neurons in several models of cerebellar dysfunction with distinct phenotypes. They develop a classifier based on parameters of single unit spike trains that seems to do an inconsistent job of predicting phenotype (though it does fairly well for tremor). The major limitation of the recording/classifier experiments is the low number of single units recorded in each model, greatly limiting statistical power. However, the authors go on to show that specific patterns of Purkinje cell stimulation cause consistent changes in interposed nucleus activity that map remarkably well onto behavioral phenotypes. Overall, I did not find the recording/classifier results to be very convincing, while the stimulation results strongly indicate that interposed nucleus firing patterns are sufficient to drive distinct behavioral phenotypes.

We thank the reviewer for their comments. We describe below how we have addressed the major concerns.

Major concerns:

(1) I don't think it's legitimate to use two 30-second samples from the same recording to train and validate the classifier. I would expect recordings from the same mouse, let alone the same unit, to be highly correlated with each other and therefore overestimate the accuracy of the classifier. How many of the recordings in the training and validation sets were the same unit recorded at two different times?

We previously published a paper wherein we measured the correlation (or variability) between units recorded from the same mouse versus units recorded from different mice (see: Van der Heijden *et al*., 2022 – iScience, PMID: 36388953). In this paper we did not find that nuclei neuron recordings from the same mouse were more correlated or similar to each other than recordings from different mice.

Upon this reviewer comment, however, we did observe strong correlations between the two 30-second samples from the same recording units. We therefore decided to no longer validate our classifier based on a training and validation sets that had overlapping units. Instead, we generated 12 training sets and 12 non-overlapping validation sets based on our entire database. We then trained 12 classifier models and ranked these based on their classification ability on the validation sets (Figure 1 – supplemental Figure 3). We found that the top two performing classifier models were the same, and used this model for the remainder of the paper.

(2) The n's are not convincing for the spike signature analyses in different phenotypic models. For example, the claim is that Pdx1Cre;Vglut2fl/fl mice have more "control" neurons than ouabain infusion mice (more severe phenotype). However, the numbers are 11/21 and 7/20, respectively. The next claim is that 9/21 dystonic neurons are less than 11/20 dystonic neurons. A z-test for proportions gives a p-value of 0.26 for the first comparison and a pvalue of 0.44 for the second. I do not think any conclusions can be drawn based on these data.

We included more cells in our analyses and found that the z-test for n the proportion of cells with the “control” and “dystonia” signature is indeed statistically significant.

(3) Since the spiking pattern does not appear to predict an ataxic phenotype and the n's are too small to draw a conclusion for the dystonic mice, I think the title is very misleading - it does not appear to be true that "Neural spiking patterns predict behavioral phenotypes...", at least in these models.

We have changed the title to: “Cerebellar nuclei cells produce distinct pathogenic spike signatures in mouse models of ataxia, dystonia, and tremor.” We feel that this new title captures the idea that we find differences between spike signatures associated with ataxia, dystonia, and tremor and that these signatures induce pathological movements.

(4) I don't think it can be concluded from the optogenetic experiments that the spike train signatures do not depend on "developmental changes, ...the effect of transgene expression, ... or drug effects outside the cerebellum." The optogenetic experiments demonstrate that modulating Purkinje cell activity is sufficient to cause changes in DCN firing patterns and phenotypes (i.e., proof-of-principle). However, they do not prove that this is why DCN firing is abnormal in each model individually.

Thank you for highlighting this section of the text. We agree that the optogenetic experiments cannot explain why the DCN is firing abnormally in each model. We have edited this section of the text to prevent this conclusion from being drawn by the readers.

Minor points:(1) It would be nice to see neural recordings in the interposed nucleus during Purkinje terminal stimulation to verify that the firing patterns observed during direct Purkinje neuron illumination are reproduced with terminal activation. This should be the case, but I'm not 100% certain it is.

We have edited the text to clarify that representative traces and analysis of interposed nucleus neurons in response to Purkinje terminal stimulation are the data in Figure 5.

(2) How does the classifier validation (Fig. 1E) compare to chance? If I understand correctly, 24/30 neurons recorded in control mice are predicted to have come from control mice (for example). This seems fairly high, but it is hard to know how impressive this is. One approach would be to repeat the analysis many (1000s) of times with each recording randomly assigned to one of the four groups and see what the distribution of "correct" predictions is for each category, which can be compared against the actual outcome.

We have now also included the proportion of spike signatures in the entire population of neurons and show that the spike signatures are enriched in each of the four groups (control, ataxia, dystonia, tremor) relative to the presence of these signatures in the population (Figure 1E).

(3) I don't think this is absolutely necessary, but do the authors have ideas about how their identified firing patterns might lead to each of these phenotypes? Are there testable hypotheses for how different phenotypes caused by their stimulation paradigms arise at a network level?

We have added some ideas about how these spike signatures might lead to their associated phenotypes to the discussion.

**Reviewer #2 (Recommendations For The Authors):**
(1) As mentioned earlier, my main concern pertains to the overall architecture and training of the classifier. Based on my reading of the methods and the documentation for the classifier model, I believe that the classifier boundaries may be biased by the unequal distribution of neurons across cerebellar disease groups (e.g., n = 29 neurons in control versus n = 19 in ataxics). As the classifier is trained to minimize the classification error across the entire sample, the actual thresholds on the parameters of interest may be influenced by the overrepresentation of neurons from control mice. To address this issue, one possible solution would be to reweight each group so that the overall weight across classes is equal. However, I suggest a better strategy might be to revise the classifier architecture altogether (as detailed below).

We have retrained the classifier model based on equal numbers of ataxic, dystonic, and tremor cells (n = 20) but we intentionally included more control cells (n = 25). We included more control cells because we assume this is the baseline status for all cerebellar neurons and wanted to avoid assigning disease signatures to healthy neurons too easily.

(2) As the authors make abundantly clear, one mouse model of disease could potentially exhibit multiple phenotypes (e.g., a mouse with both ataxia and tremor). To address this complexity, it might be more valuable to predict the probability of a certain CN recording producing specific behavioral phenotypes. In this revised approach, the output of the classifier wouldn't be a single classification (e.g., "this is an ataxic mouse") but rather the probability of a certain neural recording corresponding to ataxia-like symptoms (e.g., "the classifier suggests that this mouse has a 76% likelihood of exhibiting ataxic symptoms given this CN recording"). This modification wouldn't require additional data collection, and the exemplar disease models could still be used to train such a revised network/classifier, with each mouse model corresponding to 0% probability of observing all other behavioral phenotypes except for the specific output corresponding to the disease state (e.g., L7CreVgat-fl/fl would be 0% for all categories except ataxia, which would be trained to produce a score of 100%). This approach could enhance the validation results across other mouse models by allowing flexibility in a particular spike train parameter to produce a diverse set of phenotypes.

This is a great comment. Unfortunately, our current dataset is constrained to fully address this comment for the following reasons:

- We have a limited number of neurons on which we can train our classifier neurons. Further dividing up the groups of neurons or complicating the model limited the power of our analyses and resulted in overfitting of the model on too few neurons.

- The recording durations (30 seconds) used to train our model are likely too short to find multiple disease signatures within a single recording. We feel that the complex phenotypes are likely resulting from cells within one mouse exhibiting a mix of disease signatures (as in the *Car8wdl/wdl* mice).

We think this question would be great for a follow-up study that uses a large number of recordings from single mice to fully predict the mouse phenotype based on the population spike signatures.

To limit confusion about our classifier model, we have also altered the language of our manuscript and refer to the cells exhibiting a spike signature instead of predicting a phenotype.

However, the paper falls short in terms of the classifier model itself. The current implementation of this classifier appears to be rather weak. For instance, the crossvalidated performance on the same disease line mouse model for tremor is only 56%. While I understand that the classifier aims to simplify a high-dimensional dataset into a more manageable decision tree, its rather poor performance undermines the authors' main objectives. In a similar vein, although focusing on three primary features of spiking statistics identified by the decision tree model (CV, CV2, and median ISI) is useful for understanding the primary differences between the firing statistics of different mouse models, it results in an overly simplistic view of this complex data. The classifier and its reliance on the reduced feature set are the weakest points of the paper and could benefit from further analysis and a different classification architecture. Nevertheless, it is commendable that the authors have collected high-quality data to validate their classifier. Particularly impressive is their inclusion of data from multiple mouse models of ataxia, dystonia, and tremor, enabling a true test of the classifier's generalizability.

We intentionally simplified our parameter space from a high-dimensional dataset into a more manageable decision tree. We did this for the following reasons:

- The parameters, even though all measuring different features, are highly correlated (see Figure 1 – supplemental Figure 2). Further, we were training our dataset on a limited number of recordings. We found that including all parameters (for example using a linear model) caused overfitting of the data and poor model performance.

- Describing the spike signatures using a lower number of parameters allowed us to design optogenetic parameters that would mimic this parameter space. This would be infinitely more complex with a bigger parameter space.

We agree with the reviewer that inclusion of multiple mouse models in addition to the optogenetics experiments provide the classifier’s generalizability.

Minor Comments:(1) The blown-up CN voltage traces in Figures 5C and Supplementary Figure 2B appear more like bar plots than voltage traces on my machine.

Thank you for bringing this to our attention. We have improved the rendering of the traces.

(2) The logic in lines 224-228 is somewhat confusing. The spike train signatures are undoubtedly affected by all the factors mentioned by the authors. What, I believe, the authors intend to convey is that because changes in CN firing rates can be driven by multiple factors, it is the CN firing properties themselves that likely drive disease-specific phenotypes.

We agree that our discussion of the CN firing needs clarification. We have made the appropriate edits in the text.

**Reviewer #3 (Recommendations For The Authors):**
It's quite astounding that this can be done from single spike trains from what are almost certainly mixed populations of neurons. Could you add something to the discussion about this? Some questions that could be addressed would be would multiple simultaneous recordings additionally help classify these diseases, or would non-simultaneous recordings from the same animal be useful? Also more discussion about which cells you are likely recording from would be useful.

Thank you for this suggestion. We have added discussion about multiple recordings, simultaneous vs non-simultaneous recordings, and our thoughts on the cell population recorded in this work.

Data in figure 2 is difficult to understand - it appears that the majority of dysregulated cells in 2 ataxic models are classified as dystonia cells, not ataxic cells. This appears surprising as it seems to be at odds with earlier data from Fig 1. In my opinion, it is not discussed adequately in the Results or Discussion section.

We have added further discussion of the ataxia models represented in Figures 1 and 2.

Minor comment:The colours of the subdivisions of the bars in 2C and 3C, and the rest of the paper appear to be related to the groups in the middle (under "predicted"), but the colours are much paler in the figure than in the legend, although the colours in the bars and the legends match in the first figure (1E). Does this signify something?

These figures were remade with the same colors across the board.